# Assessing the Ecotoxicity of Eight Widely Used Antibiotics on River Microbial Communities

**DOI:** 10.3390/ijms242316960

**Published:** 2023-11-30

**Authors:** María Rosa Pino-Otín, Guillermo Lorca, Elisa Langa, Francisco Roig, Eva M. Terrado, Diego Ballestero

**Affiliations:** Faculty of Health Sciences, San Jorge University, 50830 Zaragoza, Spain; glorca@usj.es (G.L.); elanga@usj.es (E.L.); fjroig@usj.es (F.R.); evaterrado@hotmail.com (E.M.T.); dballestero@usj.es (D.B.)

**Keywords:** ecotoxicity, river microbial community, antibiotics, Biolog EcoPlates™, community-level physiological profiling

## Abstract

Global prevalence of antibiotic residues (ABX) in rivers requires ecotoxicological impact assessment. River microbial communities serve as effective bioindicators for this purpose. We quantified the effects of eight commonly used ABXs on a freshwater river microbial community using Biolog EcoPlates™, enabling the assessment of growth and physiological profile changes. Microbial community characterization involved 16S rRNA gene sequencing. The river community structure was representative of aquatic ecosystems, with the prevalence of Cyanobacteria, Proteobacteria, Actinobacteria, and Bacteroidetes. Our findings reveal that all ABXs at 100 µg/mL reduced microbial community growth and metabolic capacity, particularly for polymers, carbohydrates, carboxylic, and ketonic acids. Chloramphenicol, erythromycin, and gentamicin exhibited the highest toxicity, with chloramphenicol notably impairing the metabolism of all studied metabolite groups. At lower concentrations (1 µg/mL), some ABXs slightly enhanced growth and the capacity to metabolize substrates, such as carbohydrates, carboxylic, and ketonic acids, and amines, except for amoxicillin, which decreased the metabolic capacity across all metabolites. We explored potential correlations between physicochemical parameters and drug mechanisms to understand drug bioavailability. Acute toxicity effects at the river-detected low concentrations (ng/L) are unlikely. However, they may disrupt microbial communities in aquatic ecosystems. The utilization of a wide array of genetically characterized microbial communities, as opposed to a single species, enables a better understanding of the impact of ABXs on complex river ecosystems.

## 1. Introduction

In recent decades, the scientific literature has reported the presence of micropollutants in different aquatic ecosystems. Among these compounds, known as “emerging contaminants”, are antibiotics (ABXs), which have been widely used for the treatment of infections in humans and animals, to prevent damage by bacteria in plant cultures, and even as adjuvants in veterinary feed or food preservatives [1]. In 2013, it was estimated that 131,000 tons of ABXs were consumed worldwide, and this is expected to exceed 200,000 tons in 2030 [2].

ABXs, once consumed, become part of human and animal excretion, which constitute the main source of antibiotic release into the environment (approximately 80–90% of the total). These ABX residues can be directed to wastewater treatment plants (WWTPs) but, due their physicochemical properties, they cannot be partially or fully degraded via conventional treatments [3], allowing them to enter water bodies when wastewater is discharged into the environment [4,5,6,7]. Consequently, the amount and variety of ABXs, their metabolites, and degradation products that end up in various aquatic environmental compartments increase at a disturbing pace every year. This is the case, for example, with amoxicillin (AMO), erythromycin (ERY), and tetracycline (TC) [8,9]. ABX concentrations of around 50 µg/L [4] to 30 mg/L [10] have been reported in pharmaceutical and hospital effluents; around 0.01 to tens of μg/L [11] in municipal wastewater; and a few ng/L in surface, ground and marine waters [12]. Such values may be considered quite low; however, it must be stressed that ABX exposure times could be exceptionally long, due to their persistence in ecosystems [13,14].

The chemical properties of antibiotics significantly impact their persistence in the environment and their vulnerability to degradation processes. Consequently, degradation by-products of ABXs have been identified not only in WWT systems but also in rivers, as evidenced by the detection of TC by-products [15,16]. Antibiotics, when dissolved in water, undergo alterations, including hydrolysis, as seen in the case of chloramphenicol (CHL), penicillin (PEN), or ampicillin (AMP), photolysis (ERY and TC), sorption, and biological degradation [17,18,19]. In fact, microbial populations have a central role in the biodegradation of organic materials [20,21]. These processes are influenced by both biotic and abiotic factors, which are contingent on environmental conditions such as sunlight exposure, water temperature, the presence of microorganisms, water chemical composition, sediment properties, and organic matter content. For instance, antibiotics like ERY and TC in river water appear to initiate degradation processes within approximately four days [22,23]. These metabolites may exhibit different properties compared to the parent compounds and can contribute to the overall environmental impact.

Furthermore, potential chemical interactions among different ABXs or with other chemical compounds are poorly understood and not easy to predict.

Given that there is still no European legislation on medium-term risk assessment of emerging contaminants (and ABXs in particular), systematic ecotoxicity studies of these pollutants are imperative to establish guidelines to regulate and restrict their use and subsequent uncontrolled release into the environment.

Reported evidence reveals that ABXs harm aquatic environments, affecting all trophic levels, from microalgae [24,25,26,27] and crustaceans [28,29] to fish [29,30] and even amphibians [31]. Some degradation products have also exhibited equal or greater toxicity than the original ABXs. For example, the primary degradation products of AMO [21] and clarithromycin [32] have shown significant toxicity to fish and cyanobacteria, respectively.

However, the literature regarding the toxicity to non-target organisms of many of the antibiotics detected in rivers has significant gaps and is still inconclusive. Additionally, most studies are based on individual indicator organisms, and very few examine whole river microbial communities as endpoints [33,34,35]. Freshwater microbial communities can serve as excellent bioindicators of the impact of antibiotics (ABXs) on the riverine ecosystem [36]. They form the basis of the food web, particularly among primary producers, and also play a significant role in organic matter decomposition, thus closing nutrient cycles and participating in energy exchange, as well as pollutant degradation [37,38]. Consequently, disturbances at this level can have repercussions on all riverine communities [36,39] with unpredictable consequences for the ecological balance of the aquatic environment [40]. Therefore, to achieve a better understanding of the impact of ABXs on the aquatic environment, it is necessary to consider the effects not only on an isolated indicator organism but also on whole communities [36,41].

Until now, the few studies regarding the impact of ABXs on freshwater microbial communities have mainly focused on changes in the relative abundance of prokaryotes, and only with one or very few ABXs [42,43].

In addition, beyond acute toxicity testing, long-term studies contemplating sublethal effects are imperative to assess the impact of ABXs over long periods of time, even if their concentrations are low [33,34,35].

In the present study, we selected eight of the most consumed ABXs, detected in rivers around the world and belonging to different families with different mechanisms of action, of both narrow and wide spectrums, with the objective of quantifying their impact on a real river freshwater microbial community. For this purpose, (a) the endpoint studied was the changes in the growth and physiological profile of the bacterial community using Biolog EcoPlates™; (b) the microbial community was characterized by 16S rRNA gene sequencing; and c) the mechanism of action of each ABX was tested, as well as the values of the main physicochemical properties that might condition it (molecular weight, log Kow, and pKa), was correlated with its ability to reduce or slow down the growth of the entire community or alter its metabolic profile.

Biolog EcoPlates™ were selected for their ability to measure the capacity of a microbial community to metabolize a set of representative organic substrates [43,44,45,46,47], which indirectly provides relevant information about changes in their functional diversity [44]. Despite its great versatility, the main limitation of this approach is that it must be complemented with a taxonomic analysis of the variety of microorganisms present in the water sample. Therefore, in this study, microbial community taxonomy was also sequenced.

## 2. Results

### 2.1. 16S rRNA Gene Sequencing of River Microbial Communities

Figure 1 shows the abundance of taxa identified by DNA characterization in Krona chart plots, representing the relative abundance of microbial species detected in the samples. Input data were the average count for each taxon obtained from three studied replicates with an average of 24,818 reads. Only taxa with at least 1% total abundance are shown.

We found two predominant phyla: Cyanobacteria (56.73%) and Proteobacteria (19.77%). In addition, other phyla identified were: Actinobacteria (9.83%), Bacteroidetes (7.72%), Verrucomicrobia (4.68%), and a small proportion of Parcubacteria (1.27%).

All Cyanobacteria are oxygenated photosynthetic bacteria present in all aquatic environments, preferably in cold waters, and are an essential component of phytoplankton, which contribute to the primary production of aquatic food chains.

Proteobacteria are also very abundant in aquatic environments [48]. Based on 16S rRNA identification, this phylum is classified into three main families, all represented in our samples: Betaproteobacteria, (the most abundant, 15.49%), Alphaproteobacteria (2.92%), and a small proportion of Gammaproteobacteria (1.36%). Among the Betaproteobacteria, the only order identified was Burkholderiales, and all of these belonged to the Comamonadaceae family, among which the *Limnohabitans* genus, a relevant group of freshwater bacterioplankton [49,50], predominated (13.87%). *Hydrogenophaga*, a Gram-negative hydrogen-oxidizing bacteria, was the other genus identified within the Comamonadaceae and was much less abundant (1.62%). All Alphaproteobacteria identified belong to the genus *Paracoccus*, (Rhodobacteraceae family), a Gram-negative denitrifying bacterium with a high compound degradation capacity, useful in bioremediation [51].

All Gammaproteobacteria were Xanthomonadales and the genus identified was *Arenimonas*, a Gram-negative bacteria also found in fresh water [52].

The actinobacteria identified belonged to the order Actinomycetales, among which the genera *Rhodoluna* (4.32%) and *Cellulomonas* (3.5%) were identified. Both genera are frequent in freshwater habitats [53,54]. Actinobacteria are Gram-positive bacteria found mainly in soil and aquatic niches and play an important role as saprophytic organisms, participating in the decomposition of organic matter.

*Flavobacterium* (2.83%) and *Pseudarcicella* (1.65%) were the two genera identified among the phylum Bacteroidetes, belonging to the orders Flavobacteriales and Cytophagales, respectively. Bacteroidetes are a phylum of Gram-negative bacteria widely distributed in the environment, including in freshwater. Flavobacterium and Pseudarcicella are organotrophic aerobic bacilli that have been found to be widely distributed in freshwater [55,56]. Like most of the bacteria identified, the phylum Verrucomicrobia are Gram-negative bacteria frequently found in fresh waters; a potential role as polysaccharide degraders has been suggested for them [57]. In addition, some members of this phylum have been recently described as aerobic methanotrophs [58]. All bacteria identified belonged to the order Verrucomicrobiales and two genera were identified: *Haloferula* (2.35%) and *Luteolibacter* (1.3%).

### 2.2. Impact of Antibiotics on the Growth of River Microbial Communities

Figure 2 shows the impact of the eight selected ABXs on the growth kinetics of microbial communities obtained from river water after a seven-day incubation in Biolog EcoPlates™, measured as *AWCD*.

A clear dose-dependent response was observed in all cases(except for TC). Concentrations of 0.1 µg/mL were growth enhancers in all cases; however, with concentrations of 100 µg/mL, a clear decrease in microbial growth was observed, which was more pronounced at concentrations higher than 1000 µg/mL, except in the anomalous case of TC.

To better compare the ABX effect, Cmax (maximum achievable population density) and r (intrinsic rate of population growth) were calculated from the curves as variable responses at 1000 µg/mL (see Table 1).

Considering the Cmax values, from more effect (lower Cmax values) to less effect (higher Cmax values), the ABX that produced the greatest changes at 1000 µg/mL was CHL, and the least changes were produced by PEN, although all of them showed differences with respect to the control. At 100 µg/mL, the ABX TC presented the greatest growth inhibition but, surprisingly, promoted growth at 1000 µg/mL (see Figure 2).

On the other hand, GTM was the ABX that most affected the growth rate (lower r value) and AMO affected it the least (higher value of r), although the differences between them and the control were low.

### 2.3. Relationship between the Physicochemical Properties of Antibiotics and the Impact on Bacterial Growth

Our data have shown that the variables with the greatest weight in Cmax (Figure 3A), according to the length and perpendicularity of the arrows represented in the RDA graph, were pKa1 (pH at which the molecule under study loses its most acidic proton) and water solubility. Solubility explained 31% of the variability, while pKa1 accounted for almost 22.5%. In the case of r (slope of the AWCD line), shown in Figure 3B, the main factor was water solubility, which explained 15.16% of the variability. Following the directionality of the arrows (Figure 3A,B), we can see the comparative value and trend behavior of each ABX for each parameter; ABXs that are on the opposite side of the arrow directionality presented a lower parameter value and vice versa. As explained previously, for Cmax values, from more impact (lower Cmax values in the opposite direction to the arrow) to less impact (higher Cmax values in the same direction as the arrow), it can be seen that CHL was the ABX that produced the greatest effect and the least effect was produced by PEN.

In addition, the RDA graph allows us to see affinities in the behavior of the different ABXs according to their weight in Cmax, such that two groupings of ABXs can be identified (see ABXs grouped in red circles in Figure 3A).

In Figure 3C, we can easily observe the parameters representing the main physicochemical characteristics of the different ABXs, weighted to unify the scale. This makes it possible to observe which physicochemical characteristic is the most differential with respect to others. For example, for STM and PEN, their most remarkable parameter is water solubility; for CHL, it seems to be log Kow and, for ERY, molecular weight. The determination coefficient (goodness of fit) represents the percentage variability explained by the regression model. The percentage variability of Cmax/r was calculated for each variable using SPSS (IBM Corp. Released 2019. IBM SPSS Statistics for Windows, Version 26.0. Armonk, NY, USA: IBM Corp.).

### 2.4. Impact of Antibiotics on Community-Level Physiological Profiling

The impact of the eight ABXs on the physiological diversity of the fluvial microbiota was evaluated by studying the changes in the ability of the microbial communities to metabolize different carbon sources in the Biolog EcoPlate™ (Figure 4, Figure 5 and Figure 6).

As per previous studies [44,59,60] Biolog’s carbon sources can be grouped into five functional classes: polymers, carbohydrates, carboxylic and ketonic acids, amino acids, and amines/amides. Figure 4, Figure 5 and Figure 6 represents optical density (OD) variation after seven days’ exposure to three concentrations of the eight ABXs studied for each group of metabolites (subtracting values from the negative control). As can be seen, exposure to almost every ABX reduced the consumption of all metabolite groups, in particular with CHL (green bars). This effect was progressive over time; however, a moderate increase in the metabolism of amines, carboxylic and ketonic acids, and carbohydrates was observed for ampicillin after 48 h, as well as a major increase in all nutrient groups (except polymers) in the case of tetracycline.

## 3. Discussion

The results reported in this study revealed that the eight widely used ABXs had a significant impact on a natural river microbial community, in terms of growth kinetics and physiological profile, for concentrations ranging from 100 µg/mL. Interestingly, low concentrations (0.1 µg/mL) acted as a growth promoter.

All ABXs exhibited a dose-dependent response (except for TC), with the 1000 μg/mL dose showing the greatest reduction in microbial growth. However, both GTM and CHL displayed significant differences even at 100 μg/mL, indicating distinct mechanisms of action on microbial community growth.

To better understand these variances, growth kinetic parameters (Cmax and r) were considered, to evaluate ABX behavior.

Regarding Cmax values, from the highest to the lowest effect, ABXs were ordered as follows: CHL > ERY > GTM > STM > AMP > AMO > PEN. TC is not included in the list because it produced the greatest decrease in microbial growth at 100 µg/mL; however, it had an anomalous behavior that enhanced growth at 1000 µg/mL.

The sequence for the ability to diminish velocity growth (r) was GTM < STM < AMP < ERY < PEN < AMO; in all cases, r was lower than the control, except for CHL (but as soon as the ABX began to act, population growth practically stopped), and TC.

This impact on the metabolic profile of the microbial community was independently analyzed for each nutrient group, resulting in a general decrease in the ability to metabolize all metabolic groups after exposure to all ABXs, especially in the case of polymers at concentrations of 100 and 1000 µg/mL, with few exceptions, such as TC (Figure 4, Figure 5 and Figure 6).

As already seen in the *AWCD* curves (Figure 2), concentrations of 0.1 µg/mL were growth-promoting, with a slight increase in metabolic capacity for all metabolites after exposure to all the ABXs tested, except AMO, which caused a decrease in metabolic capacity for all metabolites at all three concentrations.

The evaluation of physicochemical properties, together with genetic sequencing of the microbial populations for the eight ABXs, allowed for a better interpretation of these results, as described below. Besides Cyanobacteria, the predominant phyla were Proteobacteria, Actinobacteria, and Bacteroidetes, a distribution highly representative of river water [61,62,63].

### 3.1. Chloramphenicol Is the Antibiotic with the Greatest Impact on River Microbiota

CHL was the ABX with the greatest impact on microbial communities and was the fastest-acting. It also caused the highest decrease in the metabolic capacity of the five metabolic groups of river microbial communities, which was significant even at 100 µg/mL. It is an ABX with a broad spectrum of action that, together with its physicochemical properties (small molecular weight, weak acid strength [64], and predominance of non-ionized forms at an aqueous pH and solubility) surely makes it highly bioavailable and effective on a great variety of microorganisms. Moreover, its partition coefficient showed a solubility in organic solvents ten-fold higher than in water, allowing it to cross biological barriers [65].

These data are consistent with the RDA graphs, which showed that log Kow and pKa1 were the parameters that most influenced the effect of CHL at Cmax (graph 3a) and r (3b). Both graphs also showed that water solubility is not a factor for CHL, as it lay in the opposite direction to the arrow marked by this physicochemical property. In graph 3c, we again see that, among all the physicochemical properties for the different ABXs, which were weighted to unify the scale, both the partition coefficient and pKa1 were the main physicochemical properties, justifying the Cmax and r of CHL.

Therefore, it could cross the envelope of the predominant Gram-negative bacteria in our samples (Proteobacteria, Actinobacteria, Bacteroidetes, Verrucomicrobia, and Parcubacteria) and move into the cytoplasmic environment until reaching its target, the 50s subunit of the bacterial ribosome. This would inhibit protein chain elongation by preventing the formation of the peptide bond [66].

Other complementary effects described for CHL that might contribute to the growth inhibition of exposed aquatic communities could be its inhibition of bacterial wall peptidoglycan and capsular polysaccharide synthesis [67,68].

The bilayer outer membrane of cyanobacteria (like that of Gram-negative bacteria) has a hydrophobic lipopolysaccharide that acts as a barrier to many drugs. However, it has been suggested that, for small non-lipophilic molecules, such as CHL, there may be a main route of entry into the cell cytoplasm of cyanobacteria, which could be the porin channels [69,70].

Nevertheless, a small proportion of microorganisms in the sample remained capable of growing in the presence of CHL, even at the highest concentration. It is common to find freshwater microorganisms that have developed resistance to CHL, including Bacteroidetes [71,72] or Actinobacteria [73].

### 3.2. Macrolides and Aminoglycosides Have a Great Impact on Microbial Growth

Despite belonging to different families (macrolides and aminoglycosides), ERY and GTM had very similar Cmax values at 1000 µg/mL, with ERY having a slightly lower value. GTM was also the ABX that most affected the microbial growth rate (it had the lowest r-value) and induced a significant reduction in growth even at 100 μg/mL, similar to CHL.

This can be seen in the RDA analysis, where both ABXs appear together in the lower left quadrant (Figure 3A in relation to Cmax) and the same quadrant, but slightly further apart (Figure 3B in relation to r). STM presented a somewhat lower effect, according to Cmax values, than ERY and GTM, but greatly affected the growth rate (it had the second lowest r value after CHL). We can also find it in the same RDA quadrant.

All three ABXs caused a decrease in the ability to metabolize all groups of metabolites at the two highest concentrations studied, especially polymers, in the case of ERY and STM, although GTM affected the metabolism of carboxylic and ketonic acids, amino acids and amines/amides slightly more, and ERY affected the metabolism of polymers more.

The mechanism of action of the two families was similar (inhibition of protein synthesis) but the ribosomal target is different. ERY acts by binding to the bacterial 50s ribosomal subunit and the aminoglycosides to the 30S s ribosomal subunit [74]. The spectrum of action is different too; STM has a broad spectrum of action, but ERY is effective mainly on Gram-positive bacteria, since it cannot pass through the cell wall of Gram-negative bacteria. For example, ERY was remarkably harmful to some groups of Actinobacteria (9.83% of our samples) [75].

GTM was usually indicated to be against Gram-negative bacteria, predominant in our samples. Previous studies [76,77] showed that several Cyanobacteria species (57% of our sample) were susceptible to GTM, which can cause cytotoxicity by the induction of reactive oxygen species, in addition to protein synthesis inhibition. Recently, Cyanobacteria were also found to be susceptible to STM, which acts as an antioxidant and disturbs photosynthesis [78].

It is worth noting that some bacteria in our samples were not affected by this ABX, as was reflected in the growth kinetic curves, and were not fully inhibited even at 1000 µg/mL, as was the case, for example, with CHL. Cyanobacterial species resistant to all three ABXs have been detected [76,79,80], suggesting that they may present intrinsic resistance to GTM and STM [81]. Communities of actinobacteria resistant to GTM have also been detected [73], so it is difficult to establish which population dynamics might explain the greater effect of these ABXs.

In addition, the physicochemical properties of these ABXs will determine their ability to cross bacterial coatings in order to reach their ribosomal target.

Erythromycin was the ABX with the highest molecular weight, and, among these three ABXs, it had the lowest water solubility (2 mg/mL). Due to its pKa (8.9), it remains non-ionized in the physiological range; however, its log Kow of −3 was low, indicating that it is not very lipophilic. The aminoglycoside gentamicin had a higher water solubility (50 mg/mL) with a lipophilicity similar to erythromycin (log Kow = −2.4) and also remains non-ionized within the physiological range. Streptomycin was the most water-soluble of the three ABXs tested (75 mg/mL), was the least lipophilic of the three (log Kow = −5.2), and also remains non-ionized within the physiological pH range.

These physicochemical data agreed with those observed in RDA Figure 3A,B, where the parameters that most influenced these three ABXs were water solubility and pKa; log Kow was not such an influential factor, as it lay in the opposite direction to the quadrant where the three ABXs under study were located. This is corroborated by Figure 3C, where the parameters that contributed variability to the parameters studied were the acidity constants and molecular weight.

It has been suggested that aminoglycosides, being positively charged molecules, can pass through the wall of Gram-negative bacteria through electrostatic interactions with the negatively charged structures of the outer membrane of these bacteria, thereby deconstructing it and allowing the ABX to enter the peptidoglycan. From there, through the cell membrane via electrical gradients, they reach the target [82,83]. ERY cannot pass through the cell wall of Gram-negative bacteria because it is a weak base and its non-ionized form will only be predominant in basic environments [84], being more effective in Gram-positive bacteria. In the case of cyanobacteria, since all these ABXs are quite large molecules, it could be that access through porins is not as direct as in the case of CHLs. This mechanism has been suggested for aminoglycosides [85].

### 3.3. Beta-Lactams Have a Minor Effect on Microbial Growth Kinetics

AMO, AMP, and PEN presented analogous changes in the growth kinetics of microbial communities, with very similar growth inhibition curves (only significant at the higher concentration), as reflected in the Cmax and r values being very close. However, AMP seemed to affect the growth rate more, as indicated by a lower r. All three ABXs caused a moderate decrease in the ability to metabolize all metabolic groups. AMP had the least effect, even producing stimulation, except in the case of polymers.

The target of action of beta-lactams is the bacterial peptidoglycan wall. They produce competitive inhibition in the last steps of wall biosynthesis, producing an alteration of bacterial integrity, leading to cell lysis [86]. For this reason, they tend to be mainly effective against Gram-positive bacterial strains that have a thick peptidoglycan wall; they are less effective against Gram-negative bacteria that have, in addition to the peptidoglycan wall, a protective outer membrane that is not the target of the ABX [87]. AMP and AMO are broad-spectrum ABXs, but PEN would only be active against Gram-positives, which were less frequent in our samples. Nevertheless, cyanobacteria seem to show different degrees of sensitivity to AMP [88], PEN [89] and AMO [90].

Moreover, these ABXs are capable of generating resistance in freshwater microorganism communities [63] and some of the taxa identified in our samples have been shown to exhibit resistance, such as Limnohabitans [91,92]. Bacteroidetes are potential hosts of several ABX resistance genes (ARGs) in fresh water [93,94] and appear to play an important role in ABX clearance [95]. There are also indications that cyanobacteria may have intrinsic resistance to beta-lactamases [96].

Taken together, this may lead to a slightly slower action of these ABXs compared with aminoglycosides or CHLs, and also a slightly lower Cmax. However, the action may be sufficient to produce a reduction in functional diversity among resistant bacteria and lead to a loss of ecological fitness [97], reflected in the slowing down of population fraction and growth (Figure 2) and a decreased ability to metabolize substrates (Figure 4).

All three beta-lactams exhibited physicochemical properties that similarly affect microbial growth kinetics, as can be seen by their location in the upper right quartile of the RDA plots (Figure 3A,B), similarities that they also shared with TC and CHL. All had similar molecular weights, and, in all cases, their log Kow (Table 1) indicated a lipophilic character, which would facilitate crossing biological membranes. At physiological pH, these ABXs would be relatively ionized due to their pKa1, which facilitates crossing bacterial coatings and, thus, makes them more bioavailable [98]. In addition, the small size of beta-lactams would justify porin channels as one of the main access routes; however, the effect of these ABXs on cyanobacteria is highly variable and perhaps dependent on differences in the amount and type of porins in the outer membrane [85]. In contrast, none of these three ABXs were very soluble in water (Table 1), which would limit the diffusion of the ABX in the periplasmic space [99]. Figure 3C shows that the most important parameters for these ABXs were the partition coefficient and acidity constants, except in the case of PEN, where solubility presented with the same importance as the partition coefficient [64].

### 3.4. Tetracycline Has an Anomalous Behavior

Our results showed that TC concentrations of 100 µg/mL produced the greatest decrease in the growth of microbial communities, compared with the other ABXs studied (Figure 2). However, at concentrations of 1000 µg/mL, as of the third day, microbial growth increased, compared with the control. This was also seen in the effect of TC on changes in the metabolic profile of microorganisms exposed to this ABX (Figure 3), since at 100 µg/mL it reduced the ability to metabolize all metabolic groups, especially polymers, whereas at 1000 µg/mL, no decrease in polymers was observed and a significant increase in the metabolic capacity of all other metabolites was detected.

This lack of dose-dependent response for TC could be due to the physicochemical properties of this compound that may lead to a decrease in its bioavailability. On the one hand, TC binds strongly to proteins and silanol groups [100] and on the other hand, this ABX tends to form coordination compounds with divalent metal ions [101] present in organic substrates. Moreover, TC is photochemically unstable, being easily photodegraded, and the resulting metabolites, mainly lactones and carboxylic acids, [102,103] are not only bactericidal [104] but could be used as substrates by bacteria, which would explain this growth-enhancing activity at higher doses.

The target of action of TC is the 30S subunit of the ribosome, to which it binds, reversibly stopping protein synthesis [105]; it has a broad spectrum of action against Gram-positive and Gram-negative bacteria. It is a poorly water-soluble molecule that acts as a weak acid when ionized at a physiological pH and is not very lipophilic, but has a low molecular weight, which probably facilitates its ability to pass through microbial coatings to reach its ribosomal target [105]. The redundancy plots in Figure 3A,B placed TC in the upper left quadrant, where it showed a higher Cmax and r value and highlighted the influence of its pKa2 and log Kow properties.

Other studies reported that TC could affect the growth of cyanobacteria [77,78,106], disturb photosynthesis [35,107] and affect protein synthesis [108]. Also, it should not be forgotten that cyanobacteria can act as reservoirs that facilitate ARG dissemination in aquatic environments [109]. Other bacteria present in our samples, like Limnohabitans, could potentially exhibit ABX resistance after months of TC exposure [110].

### 3.5. Environmental Relevance

Our results reflected that ABX concentrations within the range of 0.1 to 100 µg/mL had a considerable impact on the growth and metabolism of representative taxa of the river nekton.

The ABXs selected are among the most widely consumed worldwide, both for human and livestock use. For example, penicillin consumption in Europe (25 countries) has been estimated to be slightly higher than 230 times the defined daily dose (DDD) in 2003 [111], whereas Spain, in 2017, was the European country with the highest consumption (14.23 DDDs per 1000 inhabitants per day) [112]. In 1991, the total outpatient consumption of ERY in Finland amounted to 2.06 DDDs/1000 inhabitants per day [113], and consumption has increased since then. According to data from the EU/EEA Annual Epidemiological Report for 2020 (ESAC-Net, 2020), total antimicrobial consumption in Spain is 18.2 and in Europe, it is 15 DDDs per 1000 inhabitants on average.

Most wastewater treatment plants are not able to remove these drugs [114], which end up in sludge, as occurs with TC [8]. The same is true for AMP in liquid effluent or ERY in both sludge and liquid effluent [9]. It is not uncommon, therefore, to find levels in the ng/L range in effluents containing high concentrations of most of the ABXs we have studied herein, such as CHL [115], AMP, and ERY [116] Levels of 759 ng/L for AMO and 1000 to 5000 ng/L for ERY [117,118] have been found in this type of effluent.

ABXs pass from effluents to rivers, where they have also been widely detected. Measured TC levels in European rivers are around 50 ng/L [119], although, in some Chinese rivers, they can reach 9500 ng/L [120], which is logical, given that China is the world’s largest consumer of ABXs. In 2013, 53,800 tons were estimated to have been discharged into receiving waterways [121]. Up to 110,000 ng/L have been detected in Brazilian rivers [122]. ERY and AMP have been detected at 1149 and 184 ng/L, respectively, in rivers in Ghana [123] and AMO has been detected at up to 4630 ng/L in rivers in Brazil [124]. Some studies suggested that concentrations could reach between 0.1 and 1 µg/L at the most exposed locations of European rivers [125].

These data reflect that an acute ecotoxicity effect in rivers is not likely, since the levels of these ABXs detected in watercourses only exceed 0.1 µg/mL in extreme cases.

However, in our study, at these low concentrations, we observed an enhancement phenomenon in the ability to metabolize certain substrates for almost all ABXs, especially carbohydrates, carboxylic and ketonic acids, and amines and amides (Figure 4 at 0.1 µg/mL). On the other hand, at this same concentration, ABXs, like AMO, decreased the capacity to metabolize all substrates. This finding aligns with prior research, indicating that the metabolic functions of microbial communities are sensitive to environmental changes, and the response of specific functional flora to environmental changes is significant [126,127]. Specifically, it has been reported that environmental micropollutants can exert a significant influence on the bacterial community in surface water, even at low concentrations [128,129].

Therefore, at these low concentrations, changes occur in the growth and metabolic functions of river communities, but the extent of these changes is difficult to assess. Microbial communities play a key role in the ecological functions of the river, such as denitrification, methanogenesis, and sulfate reduction [130]. ABXs could lead to the loss or enhancement of microbial community functions, potentially affecting these ecological functions [131]. Impaired carbohydrate metabolism following exposure to AMO, for example, can impede the breakdown of organic matter in the ecosystem, thereby affecting nutrient recycling and the overall health of the ecosystem. Conversely, an increase in the capacity to metabolize these organic compounds can expedite the decomposition of organic matter within the river ecosystem, influencing the availability of resources for other organisms. This increased decomposition of organic matter may elevate the risk of eutrophication, potentially harming aquatic organisms such as fish and other vertebrates.

In these alterations, certain groups of microorganisms may derive greater benefits than others, potentially reshaping ecological interactions in the river and leading to phenomena of aquatic microecological imbalance. Sensitive individuals may succumb to the selective pressure of ABXs and their derived effects, potentially being replaced by resistant opportunists [132,133].

This microbial community imbalance must also be considered in a river environment, where a great mixture of ABXs and other drugs may be present, in addition to other emerging contaminants. All may interact with each other, creating a low-dose ABX exposure that becomes prolonged over time. Taken together, all of this could increase the risk of ABXs in freshwater environments [125].

Some studies have highlighted the risk of ABX exposure in aquatic microorganisms [34,125,134,135]; however, studies generally focused on individual indicator organisms [136] that showed great variability in their sensitivity to different ABXs [23,85]. Días [78] reported that levels in the range of 0.1 µg/mL were necessary for GTM or TC to inhibit the growth of several Cyanobacteria; however, AMO appears to be more aggressive at lower concentrations. Similar results were found by Deng et al. [42] who showed that concentrations of 0.1 µg/mL were capable of inhibiting the growth of Microcystis aeruginosa. Other authors indicated that very low concentrations of TC (300 ng/L) seemed to have a growth-enhancing effect on Cyanobacteria [137], and that concentrations higher than 50 µg/mL were necessary for TC to have any effect on different Actinomycetes [138]. Zhou [137] found that levofloxacin and oxytetracycline concentrations higher than 0.1 µg/mL affected some aquatic microbial species but not others, even at 10 µg/mL. Therefore, it is difficult to extrapolate the dose results of the effect of ABXs on individual organisms to a microbial community where effects are much more complicated, as observed, for example, by Lu [129] with the ABX ciprofloxacin, which could generate reciprocal and antagonistic interactions between cyanobacteria, other bacteria, and eukaryotes.

However, ecotoxicity studies in aquatic bacterial communities have been conducted with very few ABXs and, to our knowledge, none of the ones analyzed in this study [32,129]. Other studies did not taxonomically characterize microbial populations [43,139] or focused on other microorganisms, such as diatomaceous algae [41]. Therefore, the results for these eight ABXs may contribute toward filling the gaps regarding the impact of ABXs on river microbial communities.

Finally, a serious effect of this permanent exposure to ABXs at the levels detected is that it may contribute to local environmental ABX resistance in microorganisms. The selection of potential ABX-resistant bacteria is a phenomenon that is in full expansion [140,141,142,143] and is considered a public health threat by the WHO [144]. In fact, in effluents where ABX concentrations are particularly high, such as in hospitals, the abundance of bacteria resistant to a given ABX is very high, as has been described, for example, in the case of AMP and AMO [145].

## 4. Materials and Methods

### 4.1. Antibiotics

The ABXs employed in this study included chloramphenicol (CHL), tetracycline hydrochloride (TC), erythromycin (ERY), streptomycin sulfate (STM), and gentamicin sulfate (GTM), all of which were supplied by Acofarma (Madrid, Spain). Amoxicillin (AMO), ampicillin (AMP), and penicillin G sodium salt (PEN) were obtained from Sigma Aldrich (Darmstadt, Germany). The purity of all the products was ≥95%. Their families, physicochemical properties, and other relevant characteristics are listed in Table 1.

Stock solutions of the ABXs were prepared in distilled water at the final concentrations of 0.1, 100, and 1000 µg/mL [136].

### 4.2. River Microorganism Sampling

Freshwater samples were collected in May 2021 from the Gallego river in Zaragoza (Spain, 41°41′58.0″ N, 0°49′51.7″ W). Two samples of 1.5 L were used for the physicochemical analysis (see Appendix A) and for the community-level physiological profiling (CLPP) assay, respectively. In addition, a third 5 L sample was taken for genetic analysis.

The bottles were unsealed at the time of sampling. Prior to sampling, the water bottles were homogenized with river water five times; the sample was taken from a representative area of the river [146] and immediately taken to the laboratory. In situ parameters were also measured. The water temperature was 13 °C (Nahita thermometer); the pH was 6 (PanReac AppliChem, Barcelona, Spain); and the conductivity was 2.73 mS/cm (conductivity meter Hanna HI8733).

### 4.3. River Sample Preparation for Genetic Analysis

On the same day as sampling, according to the methodology previously described in paper [47], the 5 L of fresh water were filtered (nylon 70 µm pore diameter filter, BD Falcon^®®^,Thermo Fisher Scientific, Waltham, MA, USA) to eliminate large solid debris. Next, the 5 L of water were filtered again with ultrafiltration equipment (0.22 µm pore diameter Sterifix^®®^, B. Braun Medical S.A.U, Rubí, Spain) to retain bacteria. Once the filter became saturated, they were washed with PBS (phosphate buffered saline, pH = 7.5) and the bacterial suspensions were collected in Petri dishes. Six filters were required to filter the 5 L. Bacterial suspensions were distributed into six 10 mL Falcon tubes and centrifuged at 5000 g for 10 min at room temperature (Heraeus Biofuge Primo R centrifuge, Thermo Fisher Scientific, Waltham, MA USA). After centrifugation, supernatants were removed with Pasteur pipettes, and 10 mL of sterile water added. This process was repeated five times and the final product (six Falcon tubes with the pellet of the bacteria) was stored at −80 °C (Froilabo Trust freezer, Collégien, France) for further genetic sequencing. Samples were always handled under sterile conditions.

### 4.4. River Sample Preparation for Community-Level Physiological Profiling of Freshwater Microbes with Biolog EcoPlates™

Biolog EcoPlates™ (Biolog Inc., Hayward, CA, USA), containing 31 triplicated carbon sources for microbial metabolism, allowed us to detect changes in the physiological profiling of river microorganisms exposed to different concentrations of the ABXs tested [47,147].

In order to prepare samples for the Biolog EcoPlate™ assays, 1 L of water was taken from one of the 1.5 L bottles sampled in the river and was filtered (using the previously described 70 µm pore diameter filters) to eliminate debris. Then, the water samples were transferred to the laminar flow biological safety hood (Model MSC Advantage 1.2).

Using a multichannel pipette, 75 µL of the water sample and 75 µL of ABX solution were placed in each well of the Biolog EcoPlate™. The final pH of the solution was measured with a pH meter (Hach, Sension+ pH3) to ensure it was in a neutral pH range. Each concentration was tested in triplicate. A plate with river water filtered in the same way and without antibiotic was used as a control.

The optical density (OD) of each well was measured at 590 nm (time 0 h) using the Bio-Tek Synergy H1 microplate reader (Agilent Technologies Inc., Santa Clara, CA, USA) and Gen5™ data analysis software. Biolog EcoPlates™ were incubated at 25 °C for seven days (J.P Selecta, Barcelona, Spain). The OD was subsequently measured every 24 h for seven days.

The average well color development (*AWCD*) was calculated, as given by [148,149]:(1)$$AWCD=\overset{i=12}{\underset{i=0}{\sum }}\left(OD_{t=xi}-OD_{t=x0}\right)\;$$
where OD_t_ = x0 is the optical density for a well at the beginning of the experiment and OD_t_ = xi is the value of the optical density for the same well at any given time.

### 4.5. Taxonomical Analysis: 16S rRNA Gene Sequencing

The solution obtained in Section 4.3 was sequenced at the ADM BIOPOLIS laboratories (Parc Cientific, Universitat de Valencia, Paterna, Spain). The DNA was extracted with the High Pure PCR Template Preparation Kit (F. Hoffmann-La Roche AG, Basel, Switzterland) following the manufacturer’s instructions. The DNA concentration and quality was checked by Nanodrop 2000c (Thermo Fisher Scientific, Madrid, Spain) and agarose gel electrophoresis (1% *w*/*v* in Tris-EDTA buffer). DNA was stored at −20 °C until it was used for sequencing. A total of 50 ng of DNA was amplified, following the 16S Metagenomic Sequencing Library Illumina 15044223 B protocol (ILLUMINA, Inc., San Diego, CA, USA). Two consecutive amplification processes were carried out. In the first step, the primers designed were composed of two elements: a universal linker sequence, allowing amplicons for incorporation indexes and sequencing primers from the Nextera XT Index kit (ILLUMINA, San Diego, CA, USA), and 16S rRNA gene universal primers 341F/785R for the hypervariable region V3-V4 [141].

In the second step of the amplification process, assay amplification indexes were included. The quantification of the 16S-based libraries was carried out with a fluorimetry Quant-iT™ PicoGreen™ dsDNA Assay Kit (Thermo Fisher Scientific, Waltham, MA, USA).

The sequencing process was conducted on the MiSeq platform (Illumina) with a 300-cycle paired-end read (300 bp). Libraries were pooled before sequencing. The Bioanalyzer 2100 (Agilent) and Library Quantification Kit for Illumina (Kapa Biosciences, Wilmington, MA, USA) were used to assess the size and quantity of the pool, respectively. The PhiX Control library (v3) (Illumina) was combined with the amplicon library (expected at 20%). Data quality assessment, base calling and image analysis were performed on the MiSeq instrument (MiSeq Control Software MCS v3.1). After approximately 56 h, the sequencing data became available.

### 4.6. Bioinformatic Analysis

The quality of the reads of the FASTQ files was checked using FastQC (http://www.bioinformatics.babraham.ac.uk/projects/fastqc/, accessed on 21 September 2021). After the quality analysis, based on the results obtained, data preprocessing was carried out using Prinseq [150]. The preprocessing was performed by filtering the terminal indeterminations at both ends (N), and sequences with a mean quality (Phred quality score) lower than 26, with more than 5% of indeterminations, or sizes smaller than 150 bp were removed. Finally, bases at both ends with a quality lower than 28 were also trimmed.

The reads of the resulting files were merged, to obtain a read from overlapping paired-end Illumina sequencing reads’ forward–reverse pairs; this procedure was carried out using STICH [151].

Taxonomy was assigned using the Ribosomal Database Project (RDP) release 11 (Bacteria+Archaea combination) as the reference database [152]. Taxonomical assignation was performed with a custom-made pipeline using VSEARCH [153]. Alignment was performed, establishing high stringency filters (≥90% sequence identity, ≥90% query coverage).

### 4.7. Data Representation

A hierarchical representation of the taxonomic assignment hierarchies of microbial communities was performed using Krona [154].

XLSTAT software Addinsoft 2021 was used to build the dose–response models and calculate the standard deviations of the replicated values. Growth as a function of time was represented by *AWCD* curves, which were fitted to a logistic model [155,156], described by Equation (2) with the Excel Solver add-in (Microsoft 365), https://www.microsoft.com/microsoft-365 (accessed on 30 January 2023):(2)AWCD= Cmax/1+e^b−rt
where t is time, Cmax is the maximum achievable population density, also called carrying capacity, r is the intrinsic rate of population growth, and b is a necessary parameter for the sigmoid adjustment method, although it has no physical significance. Cmax and r were obtained for each ABX (Table 1).

The R package vegan (http://vegan.r-forge.r-project.org/, accessed on 30 January 2023) was used to perform the RDA. The radial graph was performed using normalized data for each ABX and physicochemical parameter. Normalization was carried out, considering the maximum value as one and recalibrating each value from this point.

The radial graph was constructed from the normalized physicochemical data with Excel software (office 365).

## 5. Conclusions

This study showed that eight widely consumed ABXs, present in watercourses around the world, could decrease the growth and metabolic capacity of river microbial communities, which are important as the basis of food webs and closure of carbon cycles in rivers. At ABX concentrations of 100 µg/mL, there was a reduction in microbial community growth and metabolic capacity, particularly for polymers, carbohydrates, carboxylic, and ketonic acids. CHL exhibited the greatest reduction in the capacity to metabolize all metabolites, followed by ERY and GTM. The differences detected in the toxicity of the eight ABXs likely stemmed from their distinct mechanisms of action and physicochemical properties, which may render them more or less bioavailable. A decrease in the ability to metabolize polymers or carbohydrates could potentially disrupt ecological dynamics within the river, impacting the breakdown of organic matter in the ecosystem and thereby influencing energy flow and nutrient recycling.

Yet, the impact of ABXs could already be detected at concentrations on the order of 0.1 µg/mL, where a slight increase in microbial growth and metabolism was observed (except in the case of AMO). This suggests that certain groups of microorganisms may experience benefits, leading to an imbalance in aquatic microecology that could impact the ecological functions of the river.

While acute ecotoxicity impacts are unlikely at the concentrations commonly found in rivers worldwide (except in extreme cases), the imbalance of communities at lower concentrations is not a negligible concern, especially considering the potential effects of long-term exposure to a complex mixture of ABXs and synergistic effects with other drugs and products present in the water.

This comprehensive study of a diverse range of ABXs, with different mechanisms of action on microorganisms, obtained from river waters with known taxonomy through 16S sequencing, fills the existing gap in understanding the effects of ABXs on river microbial communities and may help to better manage ABX residues to protect river ecosystems.

## Figures and Tables

**Figure 1 ijms-24-16960-f001:**
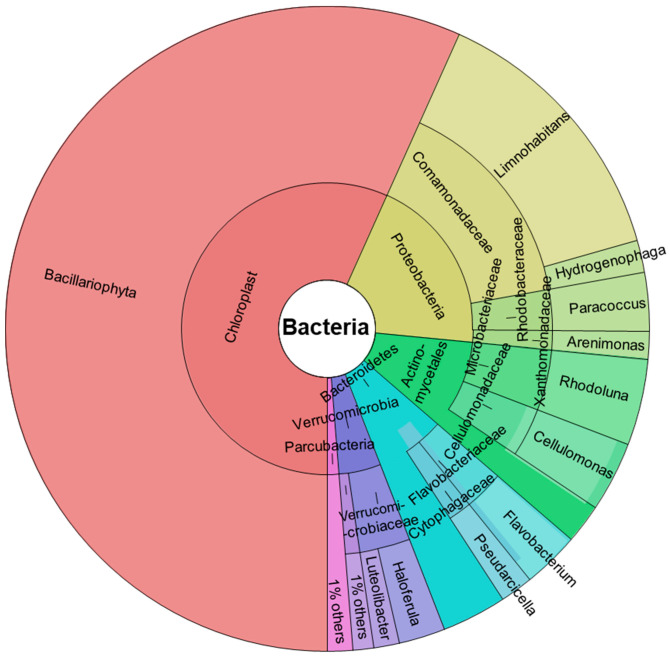
Taxonomic spectrum visualized with Krona chart of taxa showing percentage of the metagenome. Circles represent taxonomic classifications in ascending order up to the genus level (outermost circle). Only taxa with at least 1% of total abundance are shown.

**Figure 2 ijms-24-16960-f002:**
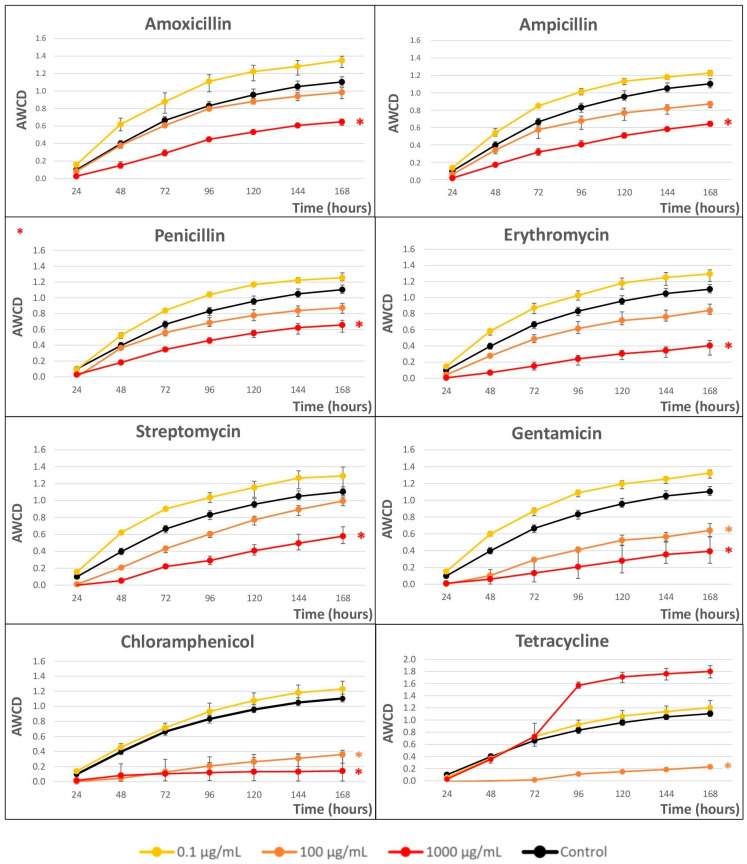
Average well color development (*AWCD*) vs. time (h) curves at three concentrations (0.1, 100, and 1000 µg/mL) for the eight antibiotics. Statistically significant differences between antibiotic exposure and control were analyzed using the Student’s *t*-test. *p*-values < 0.05 have been marked with asterisks. Error bars represent the standard deviation of the mean of three replicates (n = 3).

**Figure 3 ijms-24-16960-f003:**
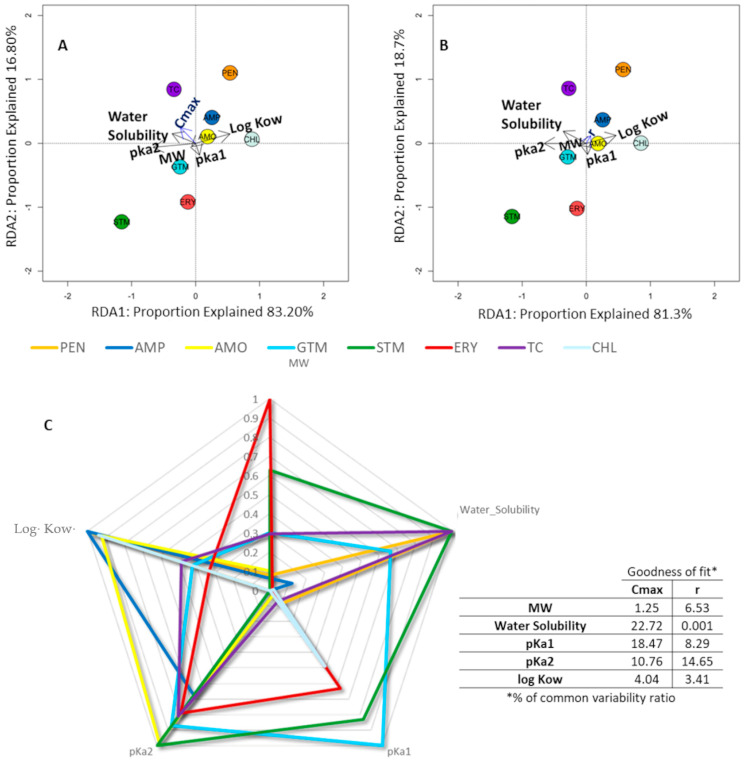
Radial graph. Redundancy analysis and physicochemical parameters representation. (**A**,**B**): redundancy analysis showing the relationship between the antibiotic and physicochemical parameters; antibiotic physicochemical properties were considered as exploratory variables and Cmax (**A**) and r (**B**) at 1000 µg/mL of antibiotic as response variables. Arrow length indicates the variance that can be explained by each parameter, with the perpendicular distance of the antibiotic to the arrow indicate the parameter’s relative importance. The groups marked in graph A are significant (*p* < 0.05). (**C**) shows the radial chart that explores each one of the studied physiochemical parameters as a dimension, with % of common variability ratio for each of the parameters.

**Figure 4 ijms-24-16960-f004:**
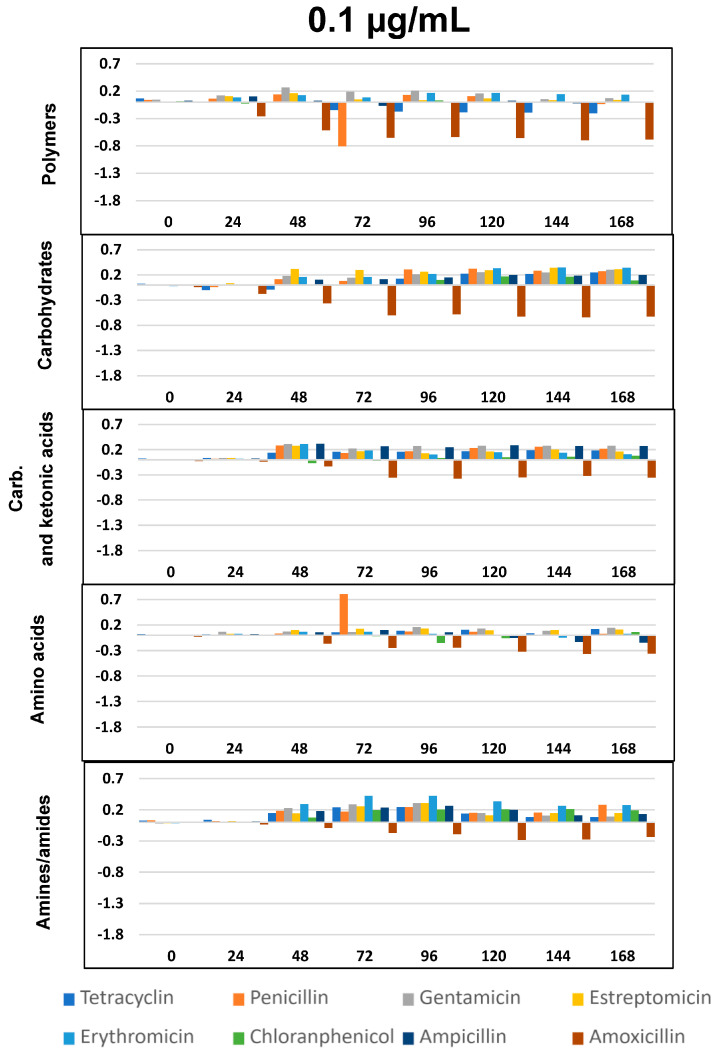
∆OD vs. time (h) curves at 0.1 µg/mL for the eight antibiotics and five selected groups of metabolites. Plotted values were obtained by subtracting those from the negative control. Statistically significant differences between antibiotic exposure and control were analyzed using the T- Student’s *t*-test. All differences with respect to the control show *p* < 0.05, except for: the chloramphenicol, ampicillin, and penicillin for polymers; tetracycline for carbohydrates; chloramphenicol for carboxylic and ketonic acids; and penicillin, chloramphenicol, and ampicillin for amino acids.

**Figure 5 ijms-24-16960-f005:**
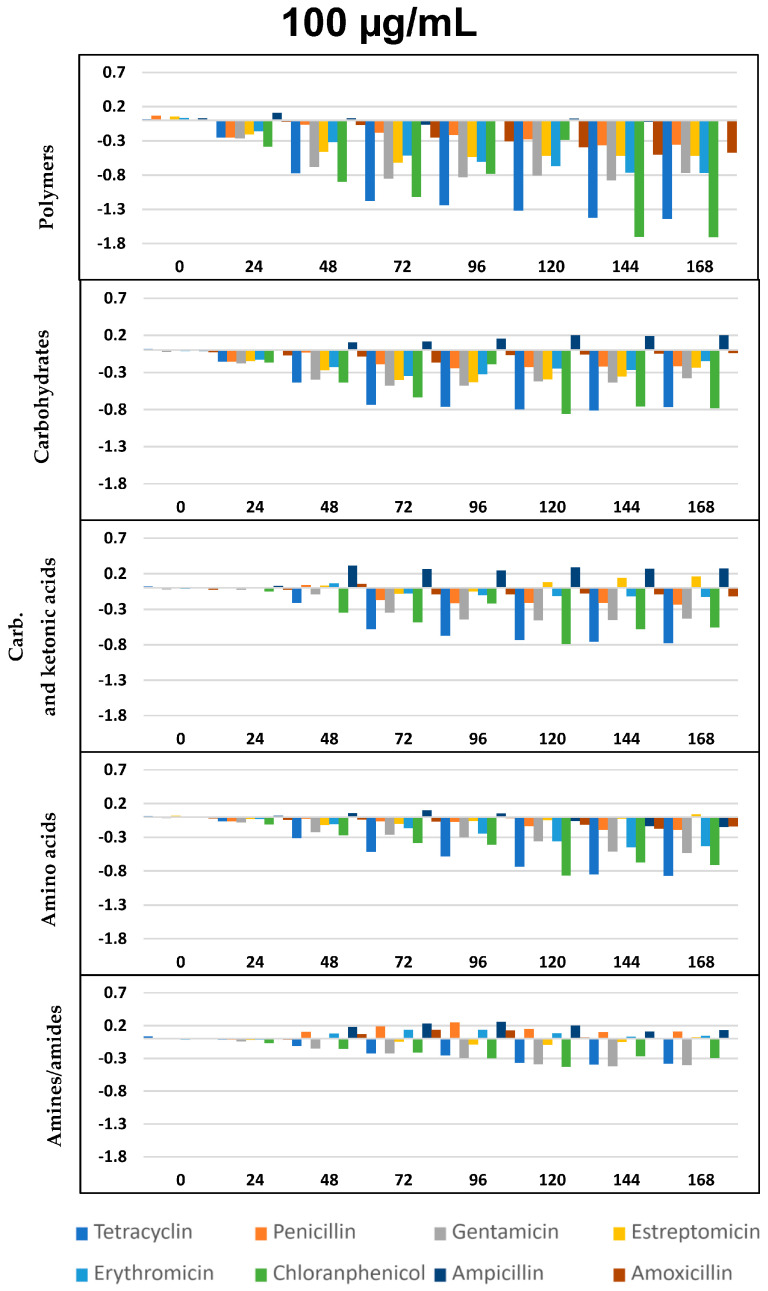
∆OD vs. time (h) curves for the eight antibiotics and five selected groups of metabolites, as in Figure 4, but at the concentration of 100 µg/mL. All differences with respect to the control show *p* < 0.05, except ampicillin for polymers and amino acids.

**Figure 6 ijms-24-16960-f006:**
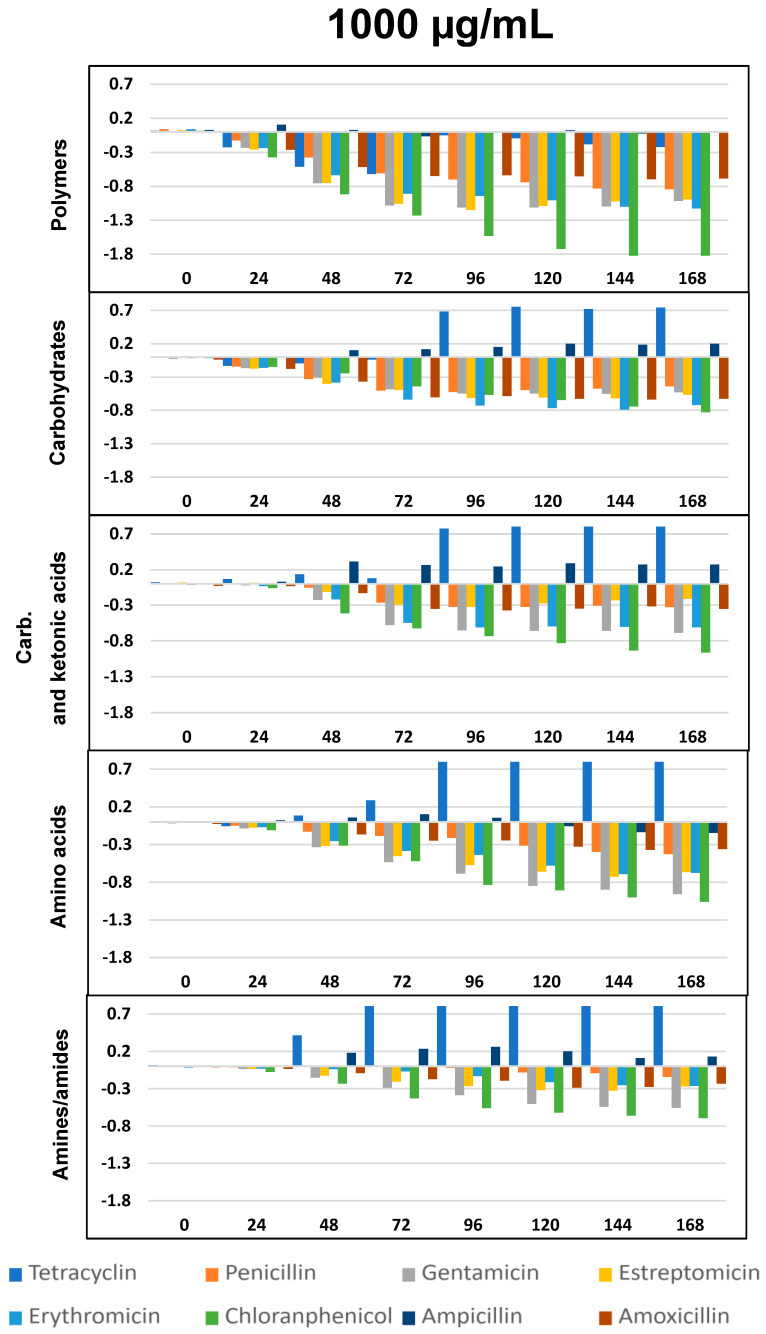
∆OD vs. time (h) curves for the eight antibiotics and five selected groups of metabolites, as in Figure 4, but at the concentration of 1000 µg/mL. All differences with respect to the control show *p* < 0.05, except ampicillin for polymers and amino acids.

**Table 1 ijms-24-16960-t001:** Properties of the eight antibiotics assayed *.

Antibiotic Name	Abbr.	Family	CAS Number	MW (g/mol)	Water Solubility (mg/mL)	pKa		Log Ko/w	1000 µg/mL	
						pKa1	pKa2		C_max_ ^(1)^	r ^(2)^
Chloramphenicol	CHL	Amphenicols	56-75-7	323.1	2.5	7.5	−2.8	1.0	0.133	0.067
Tetracycline	TC	Tetracyclines	64-75-5	444.4	75.5	3.3	9.2	−2.0	1.801	0.070
Erythromycin	ERY	Macrolides	114-07-8	733.9	2.0	8.9	8.9	−3.0	0.645	0.043
Streptomycin sulphate	STM	Aminoglycosides	3810-74-0	581.6	75.0	10.9	12.0	−5.2	0.602	0.037
Gentamycin sulphate	GTM		1405-41-0	447.6	50.0	12.6	10.1	−2.4	0.420	0.035
Amoxicillin	AMO	Beta-lactams	26787-78-0	356.4	1.0	2.9	11.7	0.9	0.645	0.043
Ampicillin	AMP		69-53-4	349.4	10.0	2.6	7.2	1.4	0.640	0.037
Penicillin G sodium salt	PEN		69-57-8	356.4	75.0	3.5	−2.8	1.2	0.650	0.042
Control									1.081	0.043

* Physicochemical properties were obtained from PubChem and ChemicalBook databases. Cmax and r were obtained from the 1000 μg/mL *AWCD* curves. ^(1)^ Maximum achievable population density (carrying capacity). ^(2)^ Intrinsic rate of population growth.

## Data Availability

Data are contained within the article or Appendix A.

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
