# Peer review of "Assessing the Ecotoxicity of Eight Widely Used Antibiotics on River Microbial Communities"

_ijms, 2023, doi:10.3390/ijms242316960_

Round 1
Reviewer 1 Report
Comments and Suggestions for Authors
The manuscript entitled “Assessing the ecotoxicity of eight widely used antibiotics on river microbial communities” by María Rosa Pino-Otín aimed to assess of ecotoxicity of selected antibiotics on naturally occurring microorganism in river ecosystems. Authors collected samples from river and conducted their research by means of commercially available test Biolog EcoPlatesTM.
Please find below my major comments:
1. In Abstract please add information about what kind of microorganisms were detected in river samples.
2. Please provide in Materials and Methods section the source of all antibiotics used in the study. Did you purified the purchased material?
3. All figures should be improved to make them readable. In current version all writings are to small.
4. Point 4.3 – how Authors developed this protocol? Are there any references in that matter?
5. Please provide an experimental protocol for evaluation of impact of antibiotics on the growth of river microbial communities.
6. Please provide an experimental protocol for metabolite average well colour development.
Author Response
REFEREE 1
The manuscript entitled “Assessing the ecotoxicity of eight widely used antibiotics on river microbial communities” by María Rosa Pino-Otín aimed to assess of ecotoxicity of selected antibiotics on naturally occurring microorganism in river ecosystems. Authors collected samples from river and conducted their research by means of commercially available test Biolog EcoPlatesTM.
Please find below my major comments:
- In Abstract please add information about what kind of microorganisms were detected in river samples.
A sentence has been added to the summary indicating the predominant groups genotyped and that the community structure detected is representative of aquatic ecosystems: Lines 11-12
- Please provide in Materials and Methods section the source of all antibiotics used in the study. Did you purified the purchased material?
The supplier of each antibiotic was indicated in Table 1. However, for the purpose of optimizing the layout of this table, the information has been moved to the Materials and Methods section, lines:619-622.
Regarding purity, the suppliers state in the product data sheet that purity is ≥95% This information has also been included in Section 4.1 of the Materials and Methods, lines 622-623).
- All figures should be improved to make them readable. In current version all writings are to small.
Of course, all figures have been improved:
Figure 1: The font size of the taxonomy and resolution has been increased.
Figure 2: The font size and resolution of the curves have been increased.
Figure 3: The font size of the antibiotic names in the legend at the base of figures 3A and 3B has been increased. Similarly, the same colours have been used to indicate each type of antibiotic for easy identification in both the bubbles of graphs 3A and 3B and the lines of figure 3C. The resolution of the figure has been enhanced. The resolution of the figure has been increased.
Figures 4, 5, and 6: The resolution has been increased in all three, along with the font size, and the size and thickness of the metabolite bars. We believe they can now be perfectly appreciated. It is worth noting that in Figure 4, the bars may appear small, but this is because the changes are minimal, and the same axis range has been maintained as in graphs 5 and 6 for comparison purposes.
- Point 4.3 – how Authors developed this protocol? Are there any references in that matter?
The procedure outlined for extracting microorganisms from a river water sample to generate pellets for DNA sequencing involves a fundamental methodology of centrifugation cycles with filters to capture bacteria, conducted under sterile conditions. We collaborated with ADM BIOPOLIS Laboratories (Parc Cientific, Universitat de Valencia), who executed the sequencing and DNA extraction processes from the bacterial pellets we prepared. Drawing upon their expertise, we optimized the sequencing process by adjusting details such as the number of centrifugation cycles and the required volume of river water, ensuring that the pellets contained an adequate amount of sample.
The procedure's validation occurred through the successful execution of initial sequencing, demonstrating high-quality results, and was first reported in our previous study (Pino-Otin et al., 2019). This standardized method for obtaining pellets of river bacteria was subsequently documented in the following publications(Pino-Otin et al., 2021a; Pino-Otin et al., 2022; Pino-Otin et al., 2021b).
Therefore, there is no standard protocol for this procedure; however, we have referenced the initial publication where it was applied, as well as in this work, in line 640-641 of Section 4.3
Pino-Otin MR, Ballestero D, Navarro E, Gonzalez-Coloma A, Val J, Mainar AM. Ecotoxicity of a novel biopesticide from Artemisia absinthium on non-target aquatic organisms. Chemosphere 2019; 216: 131-146.
Pino-Otin MR, Ballestero D, Navarro E, Mainar AM, Val J. Effects of the insecticide fipronil in freshwater model organisms and microbial and periphyton communities. Science of the Total Environment 2021a; 764.
Pino-Otin MR, Gan C, Terrado E, Sanz MA, Ballestero D, Langa E. Antibiotic properties of Satureja montana L. hydrolate in bacteria and fungus of clinical interest and its impact in non-target environmental microorganisms. Scientific Reports 2022; 12.
Pino-Otin MR, Langa E, Val J, Mainar AM, Ballestero D. Impact of citronellol on river and soil environments using non-target model organisms and natural populations. Journal of Environmental Management 2021b; 287.
- Please provide an experimental protocol for evaluation of impact of antibiotics on the growth of river microbial communities.
As before, there is no standardized protocol (apart from the instructions of use provided in the kit) in the use of BIOLOGs as they are employed for different purposes. They are frequently utilized to assess the capability of microbial communities to utilize a variety of carbon sources following exposure to drugs, based on the direct incubation of environmental samples in a Biolog EcoPlate™, as noted by (Akinwole et al., 2021; Zhen et al., 2020). This method serves as a valuable tool in distinguishing microbial communities and identifying the substrates most utilized by these communities, and it can also be helpful for comparing the effects of toxicants (Tiquia, 2010).
We followed the procedure as outlined in the 'Materials and Methods' section according to (Garland, 1997) and previous studies (Rosa Pino-Otin et al., 2019). These references have been incorporated in the line 655 of the 'Materials and Methods' section.
The Average Well Color Development (AWCD) was calculated following the methods outlined by (Garland and Mills, 1991; Tiquia, 2010), as specified in the 'Materials and Methods' section (line 669-670).
Akinwole P, Guta A, Draper M, Atkinson S. Spatio-temporal variations in the physiological profiles of streambed bacterial communities: implication of wastewater treatment plant effluents. World Journal of Microbiology & Biotechnology 2021; 37.
Zhen T, Fan W, Wang H, Cao X, Xu X. Monitoring Soil Microorganisms with Community-Level Physiological Profiles Using Biolog EcoPlates<SUP>™</SUP>in Chaohu Lakeside Wetland, East China. Eurasian Soil Science 2020; 53: 1142-1153.
Tiquia SM. Metabolic diversity of the heterotrophic microorganisms and potential link to pollution of the Rouge River. Environmental Pollution 2010; 158: 1435-1443.
Garland JL, Mills AL. CLASSIFICATION AND CHARACTERIZATION OF HETEROTROPHIC MICROBIAL COMMUNITIES ON THE BASIS OF PATTERNS OF COMMUNITY-LEVEL SOLE-CARBON-SOURCE UTILIZATION. Applied and Environmental Microbiology 1991; 57: 2351-2359.
Pino-Otin MR, Ballestero D, Navarro E, Gonzalez-Coloma A, Val J, Mainar AM. Ecotoxicity of a novel biopesticide from Artemisia absinthium on non-target aquatic organisms. Chemosphere 2019; 216: 131-146.
- Please provide an experimental protocol for metabolite average well colour development.
The experimental protocol for the average well color development of metabolites is identical to that of the growth of river microbial communities explained in question 5. The distinction lies in the data treatment: a) Instead of treating the plate as a whole, it is calculated by groups of metabolites or substrate families, as described in (lines 227-229) with values (optical density) subtracted from the negative control (Line 229). This clarification has also been added to the captions of Figures 4, 5, and 6.

Reviewer 2 Report
Comments and Suggestions for Authors
Dear Authors, the work has potential because the role that microorganisms have in nature is fundamental for the protection of our planet. Just as it is important to understand how the metabolites of the pollutants we release into the environment, such as antibiotics, transform and interact with organisms. The work must be tidied up in order to be accepted and the quality of the figures improved along with other suggestions listed below.
1) Introduction.
I suggest to better describe the degradation of antibiotics, their metabolites, their chemical nature and the interactions they have with the environment including organisms, through references.
2) Introduction.
It has recently been studied what central role microbial populations have on the biodegradation of organic materials, the authors should discuss it and cite recent works in the sector such as:
- Olivito, F.; Jagdale, P.; Oza, G. Synthesis and Biodegradation Test of a New Polyether Polyurethane Foam Produced from PEG 400, L-Lysine Ethyl Ester Diisocyanate (L-LDI) and Bis-hydroxymethyl Furan (BHMF). Toxics 2023, 11, 698. https://doi.org/10.3390/toxics11080698
- Nomadolo, N.; Dada, O.E.; Swanepoel, A.; Mokhena, T.; Muniyasamy, S. A Comparative Study on the Aerobic Biodegradation of the Biopolymer Blends of Poly(butylene succinate), Poly(butylene adipate terephthalate) and Poly(lactic acid). Polymers 2022, 14, 1894. https://doi.org/10.3390/polym14091894
3) Figure 1 is not presentable; it should be redone in higher resolution.
4) Figure 2 should be discussed more thoroughly.
5) Figure 3. Resolution is low.
6) Figure 4, 5 and 6. Resolution is low. Not presentable in this way.
7) 3. Discussion
The figures and text must be together in the Discussion chapter, not separate. Otherwise, the reader will struggle to interpret the results and the work will have a low impact.
8) The conclusions must concisely and meaningfully represent the work.
Comments on the Quality of English LanguageMinor English editing required
Author Response
REFEREE 2
Dear Authors, the work has potential because the role that microorganisms have in nature is fundamental for the protection of our planet. Just as it is important to understand how the metabolites of the pollutants we release into the environment, such as antibiotics, transform and interact with organisms. The work must be tidied up in order to be accepted and the quality of the figures improved along with other suggestions listed below.
1) Introduction.
I suggest to better describe the degradation of antibiotics, their metabolites, their chemical nature and the interactions they have with the environment including organisms, through references.
Of course, to provide a more detailed description of this matter, the following paragraphs have been added in lines 50-63 of the Introduction:
“The chemical properties of antibiotics significantly impact their persistence in the environment and their vulnerability to degradation processes. Consequently, degradation by-products of ABXs have been identified not only in WWT systems but also in rivers, as evidenced by the detection of TC by-products(Jiang et al., 2013; Zhang et al., 2023).
Antibiotics, when dissolved in water, undergo alterations, including hydrolysis, as seen in the case of chloramphenicol (CHL), penicillin (PEN), or ampicillin (AMP), photolysis (ERY and TC), sorption, and biological degradation (Kulik et al., 2023; Reis et al., 2020; Yang et al., 2021). In fact, microbial populations have a central role on the biodegradation of organic materials (Nomadolo et al., 2022; Olivito et al., 2023). These processes are influenced by both biotic and abiotic factors, contingent on environmental conditions such as sunlight exposure, water temperature, the presence of microorganisms, water chemical composition, sediment properties, and organic matter content. For instance, antibiotics like ERY and TC in river water appear to initiate degradation processes within approximately four days (Liu et al., 2019; Zhang et al., 2015). These metabolites may exhibit different properties compared to the parent compounds and can contribute to the overall environmental impact.”
And this paragraph in lines 72 to 74:
“Some degradation products have also exhibited equal or greater toxicity than the original ABX. For example, the primary degradation product of AMO (Elizalde-Velazquez et al., 2017) and clarithromycin (Baumann et al., 2015) have shown significant toxicity to fish and cyanobacteria, respectively”.
Jiang H, Zhang D, Xiao S, Geng C, Zhang X. Occurrence and sources of antibiotics and their metabolites in river water, WWTPs, and swine wastewater in Jiulongjiang River basin, south China. Environmental Science and Pollution Research 2013; 20: 9075-9083.
Zhang L, Zhu Z, Zhao M, He J, Zhang X, Hao F, et al. Occurrence, removal, emission and environment risk of 32 antibiotics and metabolites in wastewater treatment plants in Wuhu, China. Science of the Total Environment 2023; 899.
Kulik K, Lenart-Boron A, Wyrzykowska K. Impact of Antibiotic Pollution on the Bacterial Population within Surface Water with Special Focus on Mountain Rivers. Water 2023; 15.
Reis AC, Kolvenbach BA, Nunes OC, Corvini PFX. Biodegradation of antibiotics: The new resistance determinants - part I. New Biotechnology 2020; 54: 34-51.
Yang Q, Gao Y, Ke J, Show PL, Ge Y, Liu Y, et al. Antibiotics: An overview on the environmental occurrence, toxicity, degradation, and removal methods. Bioengineered 2021; 12: 7376-7416.
Liu X, Lv K, Deng C, Yu Z, Shi J, Johnson AC. Persistence and migration of tetracycline, sulfonamide, fluoroquinolone, and macrolide antibiotics in streams using a simulated hydrodynamic system. Environmental Pollution 2019; 252: 1532-1538.
Zhang H, Du M, Jiang H, Zhang D, Lin L, Yea H, et al. Occurrence, seasonal variation and removal efficiency of antibiotics and their metabolites in wastewater treatment plants, Jiulongjiang River Basin, South China. Environmental Science-Processes & Impacts 2015; 17: 225-234
Elizalde-Velazquez A, Martinez-Rodriguez H, Galar-Martinez M, Dublan-Garcia O, Islas-Flores H, Rodriguez-Flores J, et al. Effect of amoxicillin exposure on brain, gill, liver, and kidney of common carp (<i>Cyprinus carpio</i>): The role of amoxicilloic acid. Environmental Toxicology 2017; 32: 1102-1120.
Baumann M, Weiss K, Maletzki D, Schuessler W, Schudoma D, Kopf W, et al. Aquatic toxicity of the macrolide antibiotic clarithromycin and its metabolites. Chemosphere 2015; 120: 192-198.
2) Introduction.
It has recently been studied what central role microbial populations have on the biodegradation of organic materials, the authors should discuss it and cite recent works in the sector such as:
- Olivito, F.; Jagdale, P.; Oza, G. Synthesis and Biodegradation Test of a New Polyether Polyurethane Foam Produced from PEG 400, L-Lysine Ethyl Ester Diisocyanate (L-LDI) and Bis-hydroxymethyl Furan (BHMF). Toxics 2023, 11, 698. https://doi.org/10.3390/toxics11080698
- Nomadolo, N.; Dada, O.E.; Swanepoel, A.; Mokhena, T.; Muniyasamy, S. A Comparative Study on the Aerobic Biodegradation of the Biopolymer Blends of Poly(butylene succinate), Poly(butylene adipate terephthalate) and Poly(lactic acid). Polymers 2022, 14, 1894. https://doi.org/10.3390/polym14091894.
Of course, these references have been added in the comments of the previous paragraph, Lines 56-57.
3) Figure 1 is not presentable; it should be redone in higher resolution.
Of course, the font size of the taxonomy and resolution has been increased in Figure 1
4) Figure 2 should be discussed more thoroughly.
Differences in the growth curves from Figure 2 have been primarily analyzed and discussed through the kinetic parameters of the curves (Cmax and r), as they enable a more quantitative comparative analysis as elucidated in the third, fourth, and fifth paragraphs of the discussion (lines 247-255). Furthermore, this scrutiny extends to individual sections within the discussion dedicated to each antibiotic group.
In accordance with your suggestion, however..."
- A paragraph has been added to the discussion, linking it with the one discussing growth results with Cmax and r parameters: Lines 243-247.
- A sentence has been added in the paragraph 3.1 dedicated to CHL (Line 362) to connect the explanation about the effect of this specific antibiotic to the curves in Figure 2.
- A sentence has been added in section 3.2 dedicated to GTM, ERY, and STM, lines 395-396, to emphasize the ability of GTM to induce significant changes in growth even at 100 µg/mL. This capacity for growth inhibition (and substrate metabolism) is the focus of explanation in this section.
- In the paragraph 3.3 discussing the effect of AMO, AMP, and PEN on microbial communities, the first paragraph has been modified to link Cmax and r values with the behavior of inhibition curves and their significances (Lines 454-456).
5) Figure 3. Resolution is low.
Of course. The font size of the antibiotic names in the legend at the base of figures 3A and 3B has been increased. Similarly, the same colours have been used to indicate each type of antibiotic for easy identification in both the bubbles of graphs 3A and 3B and the lines of figure 3C. The resolution of the figure has been enhanced. We have a version with even higher resolution, but when inserted into the journal template, it becomes too heavy and unmanageable. The authors are available to provide it to the editor in a separate file.
6) Figure 4, 5 and 6. Resolution is low. Not presentable in this way.
Of course, the resolution has been increased in all three, along with the font size, and the size and thickness of the metabolite bars. We believe they can now be perfectly appreciated. It is worth noting that in Figure 4, the bars may appear small, but this is because the changes are minimal, and the same axis range has been maintained as in graphs 5 and 6 for comparison purposes.
7) 3. Discussion. The figures and text must be together in the Discussion chapter, not separate. Otherwise, the reader will struggle to interpret the results and the work will have a low impact.
Understood. The figures have been modified, as discussed, to enhance clarity, resolution, and size. Their placement has been adjusted, now positioned after being referenced or elucidated in the results section. Care has been taken to integrate them into the journal template to minimize gaps between figures and text, ensuring proximity to their respective descriptions or discussions.
The last three figures are presented in the Discussion section, immediately preceding their corresponding discussions. They are presented consecutively as each figure now spans a full page, allowing for detailed observation. Due to constraints within the journal's template, it is unfeasible to insert them individually amidst the text, as doing so would result in either inadequate space or disruptive page breaks.
While we believe the current arrangement enhances clarity, we remain open to any editorial suggestions or modifications.
8) The conclusions must concisely and meaningfully represent the work.
Of course, the conclusions have been completely rewritten

Reviewer 3 Report
Comments and Suggestions for Authors
Dear Editor,
I am so sorry for the late response.
It is very nice and well written manuscript. I appreciate authors for conducting such comprehensive study. Kindly, check these comments to enhance the quality of manuscript.
Specific Comments:
1. Generally, the studies dealing with microbial communities/diversity analyse data using various ecological diversity indices i.e., α (Simpson or Shannon analyses), and β (Bray-Curtis dissimilarity or Jaccard indices) diversity analyses.
2. I shall suggest analysing the relationship/effect of antibiotic/metabolites vs microbial communities must be done and data must be presented by Principal Coordinates Analyses and Canonical Analysis of Principal coordinates. It will comprehensively analyse the microbial communities.
Comments on the Quality of English LanguageGeneral Comments:
1. Line number 11: Check spacing as 16S rRNA.
2. Line number 91: Remove full stop as 16S rRNA.
3. Line number 113, 116, 119, 122, 127, 131, 272, 280, 305, 306, 309, 339, 343, 345, 357, 358, 360, 408, and 409: It is good to capitalize initial letter Gram, it is person name. It is optional.
4. Line 144: Italicize as p-value.
5. Line 153: Mention abbreviations of Cmax and r in the table legends.
Author Response
REFEREE 3
Dear Editor,
I am so sorry for the late response.
It is very nice and well written manuscript. I appreciate authors for conducting such comprehensive study. Kindly, check these comments to enhance the quality of manuscript.
Specific Comments:
- Generally, the studies dealing with microbial communities/diversity analyse data using various ecological diversity indices i.e., α(Simpson or Shannon analyses), and β (Bray-Curtis dissimilarity or Jaccard indices) diversity analyses.
We appreciate your suggestion; it is indeed a matter we value in our study when representing the impact of antibiotics on the microbial metabolic profile. In fact, in other studies, we have utilized the Shannon index for this purpose (1). However, after conducting repeated studies with BIOLOGS, we observed that there are groups of metabolites, such as amines/amides, which are significantly fewer (two in comparison to other groups containing seven or eight metabolites). Consequently, the error levels observed in this group were higher than in others, with wider confidence intervals. This parameter is indirect, relying on the only measurable aspect in these assays—the change in absorbance. As a result, we had the impression that errors were accumulating.
Furthermore, we believe that the interpretation did not significantly contribute to the objective, which was to discern changes in the growth kinetics of microbial communities and alterations in their metabolic profile. For this purpose, the Biolog EcoPlate™ has been extensively employed. For instance, in reference studies utilizing BIOLOG plates, they adopt the method of Garland and Mills (1991) and Tiquia (2010) (2-3), as we do, and analyze the values of AWCD directly without applying Shannon diversity biodiversity. We can point to other examples of similar studies with proven quality (4, 5, 6, 7 and 8).
This is why, in our recent studies (9), including the current one, we have chosen to establish a direct correlation between the effects of antibiotics and other products with the change in metabolic capacity, measured directly by the change in absorbance.
- Pino-Otin MR, Muniz S, Val J, Navarro E. Effects of 18 pharmaceuticals on the physiological diversity of edaphic microorganisms. Science of the Total Environment 2017; 595: 441-450.
- Garland JL, Mills AL. CLASSIFICATION AND CHARACTERIZATION OF HETEROTROPHIC MICROBIAL COMMUNITIES ON THE BASIS OF PATTERNS OF COMMUNITY-LEVEL SOLE-CARBON-SOURCE UTILIZATION. Applied and Environmental Microbiology 1991; 57: 2351-2359.
- Tiquia, S.M., 2010. Metabolic diversity of the heterotrophic microorganisms and potential link to pollution of the Rouge River. Environ. Pollut. 158, 1435–1443.
- Gionchetta G, Oliva F, Romani AM, Baneras L. Hydrological variations shape diversity and functional responses of streambed microbes. Science of the Total Environment 2020; 714.
- Paixao SM, Saagua MC, Tenreiro R, Anselmo AM. Assessing microbial communities for a metabolic profile similar to activated sludge. Water Environment Research 2007; 79: 536-546.
- Boivin MEY, Massieux B, Breure AM, Greve GD, Rutgers M, Admiraal W. Functional recovery of biofilm bacterial communities after copper exposure. Environmental Pollution 2006; 140: 239-246.
- Zhang L, Lyu T, Vargas CAR, Arias CA, Carvalho PN, Brix H. New insights into the effects of support matrix on the removal of organic micro-pollutants and the microbial community in constructed wetlands. Environmental Pollution 2018; 240: 699-708.
- Koner S, Chen J-S, Hsu B-M, Rathod J, Huang S-W, Chien H-Y, et al. Depth-resolved microbial diversity and functional profiles of trichloroethylene-contaminated soils for Biolog EcoPlate-based biostimulation strategy. Journal of Hazardous Materials 2022; 424.
- Pino-Otin MR, Ferrando N, Ballestero D, Langa E, Roig FJ, Terrado EM. Impact of eight widely consumed antibiotics on the growth and physiological profile of natural soil microbial communities. Chemosphere 2022; 305.
- I shall suggest analysing the relationship/effect of antibiotic/metabolites vs microbial communities must be done and data must be presented by Principal Coordinates Analyses and Canonical Analysis of Principal coordinates. It will comprehensively analyse the microbial communities.
Thank you for the suggestion. Indeed, the most effective way to assess the impact of antibiotics on the microbial community was to relate, as you proposed, the physical-chemical properties of each antibiotic (which is the differentiating factor directly linked to its mechanism of action) with the parameters we measured after exposing microbial communities to antibiotics. These parameters include overall changes in absorbance, reflecting population growth compared to the control, and changes in absorbance analyzed by metabolite groups (the five employed metabolite clusters).
To achieve this, we analyzed the kinetic behavior of microbial community growth curves after exposure to antibiotics using various parameters (Cmax, r). Finally, a Principal Coordinates Analysis was conducted (Figure 3) to examine the relationship between the effect of each antibiotic, measured by Cmax (Figure 3A) and r (Figure 3B)) and its physical-chemical characteristics (also represented in the radial chart).
In this study, we did not measure changes in the community structure after antibiotic exposure, as community sequencing was only performed on the initial sample, not after antibiotic exposure. Our objective is to assess the impact of antibiotics on the growth and metabolic profile of the community, not to study changes in the taxonomic structure of the community. Therefore, we lack data to correlate taxonomic changes with the antibiotic's effect. We can only relate what we have measured: changes in growth after exposure, measured as variations in absorbance.
The initial 16S sequencing allowed us to identify the microbial composition of the samples, providing added value to the study of the metabolic profile. This enables us to deduce how antibiotics have affected specific groups of microorganisms. The Biolog EcoPlatesTM technique was employed to gauge the impact of a substance on the entire microbial community's ability to metabolize representative organic substrates. While this technique has been used in ecotoxicity studies, interpreting results becomes more challenging without knowledge of the taxonomic composition of the microbial communities. We believe that this limitation is overcome in our study.
Comments on the Quality of English Language
General Comments:
- Line number 11: Check spacing as 16S rRNA.
- K, the space has been deleted
- Line number 91: Remove full stop as 16S rRNA.
O.K., the point is not now ( Line 112)
- Line number 113, 116, 119, 122, 127, 131, 272, 280, 305, 306, 309, 339, 343, 345, 357, 358, 360, 408, and 409: It is good to capitalize initial letter Gram, it is person name. It is optional.
O.K, gram has been replaced by Gram in all cases.
- Line 144: Italicize as p-value.
O.k, The p-value is in italics now (Line 169)
- Line 153: Mention abbreviations of Cmax and r in the table legends.
O.K. abbreviations have been explained at the bottom of the Table 1

Round 2
Reviewer 1 Report
Comments and Suggestions for Authors
Authors implemented all recommended changes. The manuscript could be published in present form.
Reviewer 2 Report
Comments and Suggestions for Authors
Dear Authors, after correcting the format of some references, the article can be accepted in this form.
Comments on the Quality of English LanguageMinor English editing required.